# Dynamic CD8^+^ T Cell Cooperation with Macrophages and Monocytes for Successful Cancer Immunotherapy

**DOI:** 10.3390/cancers14143546

**Published:** 2022-07-21

**Authors:** Anaïs Vermare, Marion V. Guérin, Elisa Peranzoni, Nadège Bercovici

**Affiliations:** 1Université Paris Cité, Institut Cochin, INSERM, CNRS, F-75014 Paris, France; anais.vermare@inserm.fr; 2Equipe Labellisée Ligue Nationale Contre le Cancer, 75013 Paris, France; 3Institut Pasteur, INSERM, 75724 Paris CEDEX 15, France; marion.guerin@pasteur.fr; 4Istituto Oncologico Veneto IOV—IRCCS, 35128 Padova, Italy; elisa.peranzoni@iov.veneto.it

**Keywords:** T cells, macrophages, monocytes, cell–cell interactions, dynamic cooperation, interferons, immunity, cancer immunotherapy, combination therapy

## Abstract

**Simple Summary:**

Innate and adaptive immunity mutually regulate one another in a dynamic fashion during immune responses. In infectious contexts, positive interactions between macrophages, monocytes and T cells are well recognized, but this is not the case in the field of cancer, where the growth of tumors disturbs the immune response. However, recent advances revealed that successful immunotherapy profoundly remodels the tumor microenvironment, promoting the activation of both T cells and myeloid cells. This review highlights the studies that hint at positive CD8^+^ T cell cooperation with monocytes and macrophages in this context, and discusses the potential mechanisms behind this.

**Abstract:**

The essential roles endorsed by macrophages and monocytes are well established in response to infections, where they contribute to launching the differentiation of specific T-lymphocytes for long-term protection. This knowledge is the result of dynamic studies that can inspire the cancer field, particularly now that cancer immunotherapies elicit some tumor regression. Indeed, immune responses to cancer have mainly been studied after tumors have escaped immune attacks. In particular, the suppressive functions of macrophages were revealed in this context, introducing an obvious bias across the literature. In this review, we will focus on the ways inwhich monocytes and macrophages cooperate with T-lymphocytes, leading to successful immune responses. We will bring together the preclinical studies that have revealed the existence of such positive cooperation in the cancer field, and we will place particular emphasis on proposing the underlying mechanisms. Finally, we will give some perspectives to decipher the functional roles of such T-cell and myeloid cell interactions in the frame of human cancer immunotherapy.

## 1. Introduction

A successful immune response is defined as an effective reaction to remove a disease-causing agent, both spatially and temporally, to prevent the illness from spreading in the body while also preventing its long-term reappearance. Hence, the first and most obvious task for immune cells is to eradicate the disease-causing agent, be it a pathogenic microorganism or a set of dysregulated cells that form a primary tumor. Surveillance to prevent the escape of the pathogen, such as a mutating virus or a tumor cell, is also crucial to inhibit the spread of the infection or metastasis. Lastly, a secured memory of the disease is necessary to avoid relapses due to dormant impaired cells.

Parallels between the activation of immune cells in response to tumors and in response to infections have always been very instructive. Historically, anti-infectious responses were associated with tumor regression, notably through the observations of William Coley, who ingeniously started to treat sarcoma patients with bacterial extracts, resulting in tumor rejection for a significant number of patients [1]. The use of compounds derived from pathogens to stimulate anti-tumor responses stems from this discovery. For example, tumor growth control and, in some cases, tumor regression can be induced by Bacillus Calmette–Guerin (BCG) or through the activation of pattern-recognition receptors with agonists of Toll-Like Receptors (TLRs) or STimulator of INterferon Genes (STING) [2,3,4,5]. On the contrary, the dysfunction of immune cells in progressing tumors has long been compared to the impaired immune response associated with chronic infections. Indeed, immune checkpoint inhibitors have arisen as an option to treat cancer from the parallel drawn with the exhausted phenotype of T-cells and the high expression of inhibitory receptors, such as Programmed cell death 1 (PD-1), in chronic infectious diseases [6,7]. Thus, there are numerous similarities in responses to infections and tumors that have inspired the design of current cancer therapies.

Of particular interest are the mechanisms of cooperation between the innate and adaptive branches of the immune system, which promote a complete and protective immune response against infections. This aspect has been difficult to address in the cancer field, as tumors are clinically detected late in the course of an anti-tumor response. In contrast, now that cancer immunotherapies are able to elicit some tumor regression, it should be possible to reinvestigate, in a dynamic fashion, how cell functions and inflammatory signals are orchestrated. In the last couple of years, we and others have documented how immunotherapy profoundly remodels the tumor microenvironment, promoting the activation of monocytes and macrophages, in addition to CD8^+^ T cells. Depletion experiments have indicated that all of these actors are necessary for optimal treatment efficacy [4,5], suggesting that positive cooperation occurs in this context. Although the use of high-throughput technologies allows for thein-depth documentation of inflammatory signals and cell subsets involved in immune responses, the relative contribution and dynamics of cellular interactions are still difficult to capture. Therefore, a parallel with the cellular interactions engaged in response to infections might help decrypt how CD8^+^ T cells and myeloid cells cooperate for successful immunotherapy.

In this review, we will illustrate how T-cells have been shown to cooperate with monocytes and macrophages during the course of immune responses. We will discuss the potential mechanisms involved in collectively building an immunity by putting in parallel the tumor mouse models that hint at such positive cooperation and lessons from anti-infectious responses. Specifically, we will point out elements that might be key in the regulation of cytokine and chemokine release, antigen presentation to CD8^+^ T cells and direct killing by myeloid cells, as well as the necessary feedback loop provided by activated T-cells. Finally, we will suggest future directions for investigating the dialog of T cells, monocytes and macrophages in human cancer immunotherapy.

## 2. Contribution of Activated Macrophages and Monocytes to T-Cell Infiltration

### 2.1. Cytokine Burst and Inflammasome-Induced Pyroptosis of Macrophages Stimulate T-Cell Recruitment

At infection sites, the activation of macrophages resulting from inflammasome formation triggers the rapid secretion of pro-inflammatory cytokines and chemokines. For example, IL-1b and IL-18 maturation in the lymph node subcapsular CD169^+^ macrophage takes place very rapidly upon infection, but, curiously, this inflammasome activation additionally induces the pyroptosis of these cells [8]. Alveolar macrophage numbers were also shown to drastically decrease at the time of lung infection [9]. In both cases, the release of pro-inflammatory factors following macrophage death triggered the local infiltration of other immune cells.

The cytokine burst caused by massive macrophage depletion is reminiscent of observations made in the context of tumors, in which regressions were induced. In those cases, the activation of the inflammasome in tumor-associated macrophages (TAMs) can result from treatment-induced tumor cell death [10], but can also be induced directly, for example, by the stimulation of the STING pathway [11]. Our team has shown that the regression of transplanted Polyoma Virus middle T antigen (PyMT) breast tumors by STING agonist treatment also relies on an initial burst of cytokines and chemokines released by TAMs, in particular, Type-I interferon (IFN) and Tumor necrosis factor α (TNFα) [5]. These cells disappeared promptly and were replaced by monocytes that took over the production of T-cell chemoattractants, notably CXCL9 and CXCL10. Similar waves of monocytes replacing TAMs were reported after cyclophosphamide treatment in a lung tumor model [12].

Aside from recruiting immune cells, the observed death of macrophages might facilitate the accumulation of monocytes in the inflamed tumor, as it eases the competition for the tissue niche [13]. However, tumors might harbor different sensitivitiesto such therapeutic modulations. We have shown, for instance, that the accumulation of TGFβ in the tumor microenvironment specifically blocks the release of type-I IFN by TAMs [14]. Despite the release of CCL2 and other cytokines underthis condition, the number of TAMs remained constant and only a weak monocyte infiltration occurred. No tumor regression was observed, suggesting that some levels of death among macrophages is necessary to initiate an optimal anti-tumor response. It would be worth investigating TAM diversity across tumor types, as recently documented [15], but also within the tumor microenvironment to understand the origin of some resistance to treatment. Indeed, TAM differentiation is driven by itsspatial distribution across tissue territories and by factors specific to the state of tumor malignancy, as recently shown by Boissonnas and colleagues [16]. It is most likely that heterogeneous responses to activation cues will arise from such TAM diversity. This would deserve further investigation in order to understand the overall resistance to inflammatory signals triggered by therapeutic interventions in some models.

Hence, the activation of the inflammasome, the death of TAMs and their concomitant release of cytokines/chemokines appears to be a central element to revive anti-tumor responses, as illustrated in Figure 1. Nevertheless, this process of activation and cell death is tightly regulated to preserve the integrity of the organism [17]. The challenge remains in understanding which micro-environmental cues are favorable to such immune activation in tumors and by which mechanisms they can be induced.

### 2.2. Additional Elements Regulating T-Cell Infiltration by Activated Macrophages and Monocytes

The amount of pro-inflammatory mediators released by innate cells constitutes a key element of regulation and must be precisely controlled to ensure a proper spatiotemporal immune response. For example, Nitric oxide (NO) plays a central role in the interplay between monocytes and T cells that occurs after infection with the Leishmania major parasite [18]. However, high levels of NO conversely limit the immune response by decreasing the inflammasome’s activity and the production of pro-inflammatory cytokines, thereby terminating immune cell recruitment [19].

In tumors, the same dichotomy can be observed. In fact, TAMs were shown to express Inducible Nitric Oxide Synthase (iNOS), for instance, after local low-dose irradiation or CpG oligodeoxynucleotides (CpG ODN) treatment in a model of pancreatic cancer, leading to the expression of adhesion molecules on the endothelium and to subsequent T cell infiltration [20,21]. In contrast, high levels of NO, which are induced by high-dose irradiation, have a negative effect on T cell tumor infiltration [21]. This illustrates another way in which, in a short time frame, TAMs might favor the recruitment of T cells with cancer therapies. In this regard, their spatial distribution near vascular regions might be key to properly activating the endothelium, suggesting that the local density of TAMs might influence the efficacy of T lymphocyte recruitment. In addition, the nature and activation status of the macrophages/monocytes might contribute to the selective recruitment of subsets of effector/memory CD8^+^ T cells.

Strikingly, there is a critical negative feedback loop at this level. Indeed, the acute cytokine burst initiated by activated innate cells is tempered by effector T cells, in addition to Treg and the metabolic quorum-sensing-like mechanism described above [19]. In a model of viral infection, it was shown that the premature death of nude mice was not caused by a high virus titer, but rather by the massive, uncontrolled production of pro-inflammatory cytokines [22]. Surprisingly, IL-10 produced by effector CD8^+^ and CD4^+^ T cells, but not by conventional Treg, was necessary to dampen the immune response and avoid tissue damage [23,24,25]. Therefore, it is worth noting that effector T cells themselves help to tame the cytokine burst from innate cells. This negative feedback loop might be interesting to consider in the context of Chimeric Antigen Receptor (CAR-T) cell therapy, in which the cytokine storms have been shown to result from the over-activation of myeloid cells [26].

## 3. Regulation of Macrophages and Monocytes Killing Activities by CD8^+^ T Cells

### 3.1. Direct Killing of Tumor Cells by Macrophages and Monocytes through NO, ROS and TNFa

The direct cytotoxic activity of monocytes and macrophages holds an important place as the first line of defense to neutralize pathogens [27]. In the context of tumors, this role is often overlooked because much attention has been paid to T-cell cytotoxicity that occurs in the later phases of the response. Nevertheless, both mouse and human studies have reported that the stimulation of macrophages by cancer therapy can result in tumor regression. For instance, in a small cohort of pancreatic ductal adenocarcinoma patients, the regression of tumors treated by chemotherapy and a CD40 agonist correlated with TAM infiltration, whereas Tumor-infiltrating lymphocyte (TIL) infiltration was surprisingly not correlated with a clinical benefit [28]. In SCID/beige mice, which lack B and T lymphocytes and Natural Killer (NK) cells, the treatment with anti-CD40 agonist induced extended survival of mice, hinting at potential myeloid tumoricidal actions [2]. In vitro, co-culture experiments performed in various studies have revealed the direct elimination of tumor cells by TAMs [4,29]. In some settings, TAMs that were isolated from regressing tumors were even better than TILs at killing tumor cells [29]. These studies underline that TAMs should be considered as effector cells in their own right, if properly activated. In fact, various therapies undertaken to reprogram TAMs might result in the re-acquisition of tumoricidal activities, as reviewed recently [30]. The ways in which TAMs kill tumor cells seem similar to what is observed in infections, mainly through the production of reactive oxygen species (ROS), NO and TNFα [4,31,32,33].

Another specific function of macrophages is phagocytosis. It was shown that monocyte-derived macrophages (defined as Ly6C^low^ CD64^+^ FTL1^+^) perform the phagocytosis of tumor cells in vivo compared with the resident-derived macrophages [12]. This effector function could be facilitated upon antibody-mediated treatment, as the assessment of antibody-dependent phagocytosis induced by anti-CD20 therapy revealed a dominant role of TAMs in tumor elimination [34]. This approach highlights the importance of combining CD8^+^ T cell stimulation with a reinforcement of macrophage activity that could be achieved, for instance, by the use of anti-CD47 blocking antibodies. CD47 overexpression is used by tumor cells as a “do not eat me signal” that inhibits macrophage phagocytosis after binding to Signal regulatory protein α (SIRPα). This effect can be counteracted by molecules that interfere with CD47/SIRPα binding, as demonstrated by several clinical trials combining CD47 inhibitors with immune checkpoint blockers, either by monoclonal Abs combinations or by bispecific compounds [35,36,37].

Nevertheless, many questions deserve to be addressed to gain further insight into the relative contribution of TAMs and monocytes to direct tumor cell killing in vivo, as both withhold capacities to kill tumor cells. Another open question is how far in space can the killing of tumor cells occur spatially. Phagocytosis is spatially restricted and the toxicity mediated by unstable derivatives, such as Reactive oxygen species (ROS), is probably local and cannot spread in the tumor microenvironment, as previously suggested [38]. Moreover, and above all, the signals necessary to regulate the tumoricidal activity of macrophages and monocytes, and how this activity takes place in time and space with regards to that of T-cells, deserves to be studied in depth.

### 3.2. Tumoricidal TAMs Receive Activating Signals from TILs

Favorable microenvironmental cues seem to be required for the emergence of tumoricidal activity in TAMs. It seems that macrophages need several signals, such as IFNs and TLR signaling [39,40], to kill tumor cells, as illustrated with Lipopolysaccharides (LPS)/IFNγ-activated macrophages (Figure 2 and Appendix A). The activated macrophages and tumor cells were imaged every 5 min with a wide-field fluorescence microscope over 15 h to visualize the killing of tumor cells. As shown in this example, a macrophage in red formed a long-lasting conjugate with a tumor cell for the first couple of hours, before death of the tumor cell was detected in the last hour of recording. 

In the context of microbial infections, the IFNγ produced by T lymphocytes activates innate cells [41], which are found to be necessary for complete pathogen clearance through confinement and direct killing. Interestingly, decades ago, TAMs were shown to gain tumor-cell-killing activity through interactions with lymphocytes [42]. Recently, T cell-derived IFNγ was found to have profound effects on the reactivation of the whole tumor microenvironment [43,44], likely including the induction of tumoricidal activity in myeloid cells. In line with this, Hollenbaugh and colleagues showed that the ability of host immune cells to respond to the T-cell-derived IFNγ was crucial to inducing acute tumor rejection after adoptive T-cell transfer in the EG7 tumor model [45]. This anti-tumor response was associated with high numbers of infiltrating myeloid CD11b^+^ and Gr-1^+^ cells, and relied on NO production by these cells.

In another mouse model, we demonstrated that tumors start to shrink after vaccination and IFNα injection [4], at a time where CD11b^+^ cells were found to be cytotoxic against tumor cells ex vivo. A few days before, even though CD8^+^ TILs constituted less than 5% of live cells in the tumor, they were in close contact with the activated MHCII+ F4/80^+^ myeloid cells in situ. In addition, the depletion of CD8^+^ T cells or treatment of IFNγ-KO mice led to the decreased activation of monocytes and TAMs, and no tumor regression. This observation supports the idea that IFNγ-producing TILs help TAMs in tumor cell killing [4,46]. In addition, it is important to consider that both CD8^+^ and CD4^+^ T cell types may promote such activation of TAMs, as previously shown [47].

Interestingly enough, many combinations of therapeutic treatments could favor cooperation between TILs and TAMs at the level of tumor cell killing, even if not designed with this precise aim [30]. Among the few authors who have, however, sought to provoke TAM activation by TILs, Chmielewski and colleagues treated mouse colon cancer with engineered CAR-T cells that secreted IL-12p70 when engaged with a tumor cell. In their experiments, macrophages were locally activated, as shown by their up-regulation of CD80 and CD86 and by TNFα production. When macrophages were depleted, no tumor cell elimination was observed, indicating that the IL-12-induced elimination of tumor cells relied on killing by TAMs [48].

Altogether, these various observations support the concept that, besides the activation of cytotoxic effectors, such as TILs, TAM activation also appears to be crucial to eliminate tumor cells. This is particularly appealing, as TAMs’ killing mechanisms differ to those of TILs, may occur with different kinetics and, most importantly, are antigen-independent. As a whole, this cooperation might not only be additive, but also synergistic, and lead to complete tumor eradication.

## 4. Local Reactivation of T Cells by Macrophage Antigen Presentation

Another important way in which macrophages, monocytes and T cells could cooperate is through the presentation of tumor-associated antigens. Even though dendritic cells (DCs) are evidently superior at priming T cells, this does not rule out the possible role and importance of macrophages in antigen presentation to T cells. However, the role of macrophages as T cell activators is still controversial. In both infectious and tumor fields, macrophages have long been considered to have almost no tie with adaptive immunity, and DCs are usually given all the credit for presenting antigens to T cells. In the context of immunotherapy, however, the nature of the cells able to stimulate anti-tumor T cells deserves to be looked at more closely.

### 4.1. Cross-Presentation by Macrophages in Lymphoid Organs Stimulates a First Wave of Effector CD8^+^ T Cells

Mouse studies have revealed that subcategories of macrophages are able to ingest antigens and cross-prime T cells in some situations. Most importantly, macrophages and DCs were shown to activate T cells at different times of the response and perform slightly different roles. For example, in the spleen, Red Pulp Macrophages (RPMs) were shown to activate Kb/OVA-specific OT-I CD8^+^ cells, in support with cDC1, upon the injection of soluble OVA antigen [49]. Both cell types had their own efficient T-cell-priming time window. Indeed, RPMs promoted strong proliferation and cytotoxicity of OT-I cells when isolated 3h after OVA uptake in vivo, but not after 18h. On the contrary, the OVA load on cDC1 was always lower than that on RPMs and induced OT-I activation only when isolated at late time points. Importantly, this reveals that T cell priming by macrophages and DCs shift in time and complement one another. The authors further demonstrated that RPMs are supportive of a first Cytotoxic T lymphocytes (CTL) antiviral response, while a second massive wave of effector cells is later activated through cDC1 to complete viral clearance and induce memory CTLs. RPMs have a supportive, yet not essential, role in that matter.

Evidence for the early presentation of antigen by macrophages in lymphoid structures also exists in the context of tumors. For example, sinus CD169^+^ macrophages in draining lymph nodes form conjugates with naive T cells very early after the injection of apoptotic tumor cells [50], and the transient depletion of these macrophages at the time of immunization decreases the proliferation of antigen-specific CD8^+^ T cells and prevents the rejection of tumor cells upon re-challenge. These results suggest that the cross-priming of naive CD8^+^ T cells in the lymph node through intake of dead cells relies on the presence of CD169^+^ macrophages. In support of this, it was reported that a high density of CD169^+^ cells in human tumor-draining lymph nodes correlated with a smaller tumor size and, overall, earlier clinical stages in breast cancer patients [51]. 

Together, these data hint at a positive role of macrophages in the development of antitumor activity in lymphoid organs.

### 4.2. TAMs are Abundant and Efficient Antigen-Ingesting Cells that Interact with T Cells at the Tumor Site

Various studies have already indicated that TAMs qualify for the local reactivation of T cells in the tumor microenvironment. Broz and colleagues showed, in a variety of mouse models, that most tumor antigen uptake was performed by F4/80^hi^ TAMs [52]. It was further shown that macrophages and Ly6C^hi^ inflammatory monocytes are the most efficient myeloid cell subtypes to ingest antigens at the tumor site [53,54], considering that antigen uptake by cDC1 was delayed in comparison with macrophages and monocytes. Nonetheless, migratory DCs were much better at presenting tumor antigens [54] in the tumor-draining lymph nodes. 

At the tumor site, however, these TAMs and Ly6C^hi^ monocytes were found to be more abundant and dispersed throughout the tumor mass [53,54], and therefore more likely to meet with the numerous T cells. Furthermore, TAMs that had engulfed antigenic fragments interacted with T cells for long periods of time and could indeed cross-present tumor-antigens to CD8^+^ T cells through MHC-I/TCR signaling in vitro [53]. Interestingly, these TAMs efficiently activated naive T cells, but had rather inhibitory effects on previously activated CD8^+^ T cells, unless supplemented with IL-2 or TLR signals. Likewise, CD8^+^ T cells help shape the differentiation of TAMs, as Colony stimulating factor 1 (CSF1) release by exhausted CD8^+^ T cells was shown to favor antigen-presentation transcriptional programs in tumor-infiltrating monocytes [55].

These observations reveal two important aspects of CD8^+^ T cell activation in tumors. First, macrophages can cross-present antigens to CD8^+^ T cells, but in the absence of additional signals, this causes T cells to undergo exhaustion [55], a frequently observed state in tumors [56]. Secondly, these results suggest that TAM and CD8^+^ T cell co-activation should be investigated in the context of tumor regression, in which the abundant presence of pro-inflammatory signals, mimicked by TLR signals [53], could favor effector T cell re-activation. Of particular interest is the type of T-cell differentiation program initiated in such regressing tumors. For instance, monocyte-derived cells appear to promote resident memory T cell differentiation during the course of infections [57]. Finally, the reactivation of CD8^+^ T cells might also further activate TAMs through the secretion of IFNγ [4], suggesting bidirectional cooperation between these cells.

As a whole, macrophages were already shown to be remarkable at up-taking antigens in both secondary lymphoid organs and at the tumor site. This was, however, shown separately, hinting out for a more comprehensive study of T cell activation by TAMs in both the draining lymph node and at the tumor site. Importantly, some human tumors harbor tertiary lymphoid structures in which naive T cells can also be primed [58]. The analysis of the role of TAMs in these structures might reveal an additional antigen-presentation function of macrophages. Needless to say, antigenpresentation on MHC II and the activation of T helper CD4^+^ is another aspect of myeloid cell-T cell cooperation that should be taken into account [59,60].

Overall, the rare studies that aim at revealing the actors and events involved in effector T cells’ activation and differentiation at the tumor site would be essential to improve the efficacy of anti-cancer immunotherapy.

## 5. TIL and TAM Cooperation in the Frame of Human Cancer Immunotherapy

In human cancers, TAMs are among the most abundant cell types at the tumor site and are often associated with an immunosuppressive, pro-tumoral microenvironment, and with progressive disease and overall poor prognosis. However, the presence and abundance of specific M1-like TAM subsets that express markers related to cytotoxicity, antigen presentation and antibody-dependent cell cytotoxicity (such as NOS2, TNFα, HLA-DR, CD86 and CD16) are associated with good prognosis in different solid tumors, as reviewed in [61]. Interestingly, a similar phenotype of TAMs and the expression of myeloid genes involved in antigen processing and presentation, chemoattraction or IFN signaling has been recently observed in baseline tumor biopsies of cancer patients responding to immune checkpoint inhibitors [62,63,64]. In these reports, the presence of anti-tumor TAMs was coupled with the infiltration of cytotoxic CD8^+^ T cells and activated CD4^+^ T cells, as well as with inflammatory cytokines and chemokines. The single-cell analysis of the baseline tumor microenvironment of patients bearing different types of solid tumors has revealed that pro-inflammatory TAMs are enriched in patients with good prognosis and in responders to immune checkpoint inhibitors, and are also the main producers of the CXCL9, CXCL10 and CXCL11 lymphocyte chemoattractants [65,66,67,68,69,70]. Other reports correlate the presence of Programmed death Ligand 1 (PD-L1^+^) TAMs with the clinical benefit of immunotherapy, indicating that this phenotype might arise from (and likely contribute to) an IFNγ-rich anti-tumor microenvironment (reviewed in [62,69]. In accordance with this potential TIL and TAM cooperation, some authors have reported that monocytes with a phenotype similar to anti-tumor TAMs, i.e., expressing CD16, HLA-DR, PD-L1 and CD86, can be found at the baseline and during immunotherapy in the blood of cancer patients responding to immune checkpoint inhibitors [66,71,72,73], suggesting that these cells might be the precursors of M1-like TAMs found at the tumor site.

Given the difficulty in obtaining tumor biopsies during or after immunotherapeutic treatments, we have very little information on the dynamic interactions between TAMs and T-lymphocytes in regressing tumors. However, clinical trials, including neo-adjuvant therapy with immune checkpoint inhibitors or the collection of post-treatment biopsies, have revealed that, similar to T cells, the phenotype of monocytes and macrophages is modified upon immunotherapy, primarily through the action of interferons and other pro-inflammatory cytokines. For instance, after immunotherapy, macrophages were found to upregulate PD-L1, various genes related to antigen processing and presentation, as well as genes involved in phagocytosis and IFN-dependent CXCL9, CXCL10 and CXCL11 chemokines [66,69,74,75,76,77,78]. In some of these studies, co-stimulatory molecules (CD80, CD86 and ICOSLG) and chemokine (CXCL9)-related interactions are predicted between T cells and monocytes/TAMs in patients responding to immunotherapy. In contrast, inhibitory interactions (involving LILRB4, TGFB1, GRN or VEGFB) between these cell types are more frequently observed in non-responders [69,79]. In addition to the molecules involved, the localization of these interactions in the microenvironment might also be relevant to understanding their functional consequences: progressing tumors with an immune-excluded infiltrate are rich in stromal interactions between M2-like TAMs and TILs [80,81], while CD28-activating interactions are observed between intraepithelial TILs and TAMs in tumor patients displaying a cytotoxic immune response and good survival [82].

It is nonetheless important to note that the activation of the microenvironment not only leads to the orchestrated elimination of tumor cells, but also to the simultaneous induction of negative feedback mechanisms promoting progressive immunosuppression to terminate the immune response. This can eventually result in resistance in patients when the negative feedback mechanisms outbalance the inflammatory signals before the tumor is eradicated.

## 6. Concluding Remarks and Future Directions

For a long time, kinetics studies performed during the course of primary infections have revealed key connections between the innate and adaptive arms of our immune system. In particular, both myeloid cells and T cells hold an important place, as they are both supportive and killer cells, depending on the stage of the immune response. The few recent investigations performed in the frame of regressing tumors indicate that T-cells recruited upon immunotherapy need to interact in various ways with pro-inflammatory macrophages and monocytes to build an optimal antitumor response. As stated above, this corresponds to a dynamic dialogue to alert, attract and reactivate each other to effectively eliminate tumor cells. Therefore, understanding the role of these different actors and how deeply interconnected their functions during tumor regressionare might have a major impact on the design of upcoming combination therapies and the identification of predictive biomarkers.

## Figures and Tables

**Figure 1 cancers-14-03546-f001:**
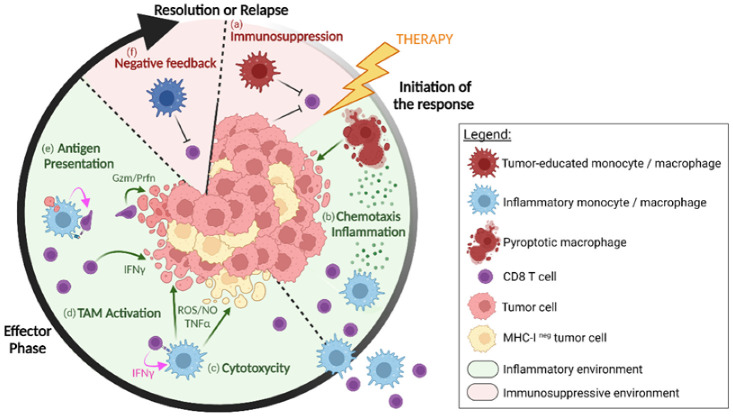
The different levels of cooperation between monocytes/macrophages and CD8^+^ T cells in regressing tumors after immunotherapy. In progressing tumors, tumor-educated macrophages contribute to inhibiting CD8^+^ T cell activities (**a**). Upon immunotherapy, macrophages release inflammatory cytokines and chemokines (**b**), concomitant with macrophage pyroptosis. It attracts and guides new myeloid cells and CD8^+^ T cells to infiltrate the inflamed tumor. Monocytes/macrophages can also kill tumor cells (**c**), following activation by IFN-γ-producing CD8^+^ T cells (**d**), and some subsets might locally reactivate the CD8^+^ T cells through antigen cross-presentation (**e**), increasing the probability of tumor cell killing. As the tumor regresses, a natural negative feedback loop (**f**), that goes along with the activation of effector cells, progressively terminates the immune response. Created with BioRender.com (https://biorender.com/ accessed on 10 June 2022).

**Figure 2 cancers-14-03546-f002:**
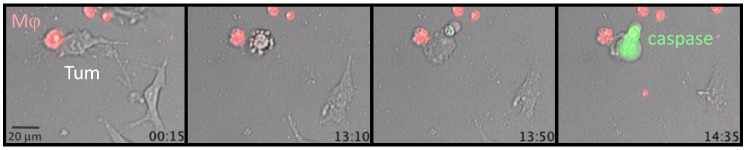
Tumor cell killing by activated macrophages. LPS/IFNγ−activated BMDM (Mϕ, Red) was cultured with tumor cells and the cleaved caspase 3 fluorescent probe (green) to visualize death by dynamic imaging. Snapshots of a dynamic imaging recording for a 14.35 h period with a widefield fluorescence microscope (objective 20×). Bottom right of each image: time (in hours) after the beginning of the recording. The corresponding movie can be visualized in the Appendix A (frame interval is 5 min).

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
