# Peer review of "Dynamic CD8+ T Cell Cooperation with Macrophages and Monocytes for Successful Cancer Immunotherapy"

_cancers, 2022, doi:10.3390/cancers14143546_

Round 1
Reviewer 1 Report
Dear Authors,
Dynamic CD8+ T cell cooperation with macrophages and mon- 2 ocytes for successful cancer immunotherapy by Vermare et al is very interesting. However, to improv ethe quality of the manuscript additional data/information is required.
Comment 1. In Fig 1 the write-up/legends are visible and readable.
Comment 2. Fig 2. The authors could give the explanation for 7 hr ,14:00 hr and 14:30 hs will be helpful for the readers to understand.
Comment 3. Figure 2. The authors could show the final minute video clip for activated macrophage and tumor cell interaction will greatly improve the manuscript quality.
Author Response
Dear Reviewer,
We thank you for your comments and suggestions. We have revised the manuscript accordingly.
Comment 1. In Fig 1 the write-up/legends are visible and readable.
We have changed the police of the legends on the Figure 1 to make it more readable (please see Figure 1 on page 3).
Comment 2. Fig 2. The authors could give the explanation for 7 hr ,14:00 hr and 14:30 hs will be helpful for the readers to understand.
Indeed, following your comment, we have revised the legend of the Figure 2 (page 5 Line 233) to precise that the time indicated on the bottom right of each image corresponds to the time, in hours, after the beginning of the recording. We have also selected 4 images, to provide the most representative snapshots of the movie.
Comment 3. Figure 2. The authors could show the final minute video clip for activated macrophage and tumor cell interaction will greatly improve the manuscript quality.
We thank you for this suggestion. We have added the movie as “supplementary Movie 1” for the reader to see the entire sequence (page 5, line 222).
Reviewer 2 Report
This is a comprehensive, well written and updated review on an interesting topic in cancer immunotherapy: how to orchestrate cooperation between T cells and macrophages/monocytes to completely eradicate tumor cells. TAM reprogramming for this purpose deserves more in-depth research, taking into account the relevant therapeutic implications for cancer patients.
The potential of anti-CD47 antibodies is addressed at different points; perhaps it should be of interest to include a reference to anti-CD47 x anti-PD-L1/anti-PD1 bispecific antibodies (already in clinical trials) or similar bifunctional constructs able to simultaneously activate both branches of immunity against tumors.
Minor comments
-In the abstract, semicolons are in the place of commas
-Keywors are missing
-There are gratuitous hyphens scattered throughout the text (v.g.: fi-nally, tu-mors, re-sponse, macro-phage)
-Line 49: “specific pathogens recognition receptors” perhaps refer to “pattern recognition receptors” (PRR)?
-Figure 1 Legend: “marcophages”, IFN-@-producing CD8+ T cells
Author Response
Dear Reviewer,
We thank you for your comments and suggestions. We have revised the manuscript accordingly.
Comment 1. In Fig 1 the The potential of anti-CD47 antibodies is addressed at different points; perhaps it should be of interest to include a reference to anti-CD47 x anti-PD-L1/anti-PD1 bispecific antibodies (already in clinical trials) or similar bifunctional constructs able to simultaneously activate both branches of immunity against tumors.
We thank you for this suggestion. We have added the following sentence, with three references “This approach highlights the importance of combining CD8+ T cell stimulation with a reinforcement of macrophage activity that could be achieved for instance with the use of anti-CD47 blocking antibodies. CD47 overexpression is used by tumor cells as a “don’t eat me signal” that inhibits macrophage phagocytosis after binding to SIRPα. This effect can be counteracted by molecules that interfere with CD47/SIRPα binding, as demonstrated by several clinical trials combining CD47 inhibitors with immune checkpoint blockers, either by monoclonal Abs combinations or by bispecific compounds [35–37]”. (page 6, line 200).
Minor comment 1: In the abstract, semicolons are in the place of commas
We apologize but it seems that these typo errors were introduced after submission of our original version of the manuscript, most probably during formatting. The revised version has been corrected.
Minor comment 2: -Keywors are missing
The revised version now includes the keywords (that were in the original version of our manuscript).
Minor comment 3: There are gratuitous hyphens scattered throughout the text (v.g.: fi-nally, tu-mors, re-sponse, macro-phage)
We apologize again but as mentioned above, it was not in the original version we have submitted. The revised version is now corrected.
Minor comment 4: -Line 49: “specific pathogens recognition receptors” perhaps refer to “pattern recognition receptors” (PRR)?
We have modified the text accordingly (page 2, line 52)
Minor comment 5: -Figure 1 Legend: “marcophages”, IFN-@-producing CD8+ T cells
These typo errors were not in the initial version but the revised version is correct.
Round 2
Reviewer 1 Report
Now the quality of the manuscript is improved.